# The Usability of an Online Tool to Promote the Use of Evidence-Based Smoking Cessation Interventions

**DOI:** 10.3390/ijerph182010836

**Published:** 2021-10-15

**Authors:** Daniëlle N. Zijlstra, Catherine A. W. Bolman, Jean W. M. Muris, Hein de Vries

**Affiliations:** 1Department of Health Promotion, Maastricht University/CAPHRI, Peter Debyeplein 1, 6229 HA Maastricht, The Netherlands; hein.devries@maastrichtuniversity.nl; 2Department of Psychology, Open University of the Netherlands, Valkenburgerweg 177, 6419 AT Heerlen, The Netherlands; catherine.bolman@ou.nl; 3Department of General Practice, Maastricht University/CAPHRI, Peter Debyeplein 1, 6229 HA Maastricht, The Netherlands; jean.muris@maastrichtuniversity.nl

**Keywords:** smoking cessation, evidence-based interventions, decision aid, usability, decision-making

## Abstract

To increase usage of evidence-based smoking cessation interventions (EBSCIs) among smokers, an online decision aid (DA) was developed. The aims of this study were (1) to conduct a usability evaluation; (2) to conduct a program evaluation and evaluate decisional conflict after using the DA and (3) to determine the possible change in the intention to use EBSCIs before and directly after reviewing the DA. A cross-sectional study was carried out in September 2020 by recruiting smokers via the Internet (*n* = 497). Chi-squared tests and *t*-tests were conducted to test the differences between smokers who differed in the perceived usability of the DA on the program evaluation and in decisional conflict. The possible changes in intention to use EBSCIs during a cessation attempt before and after reviewing the DA were tested using *t*-tests, McNemar’s test and χ^2^ analysis. The participants evaluated the usability of the DA as moderate (MU; *n* = 393, 79.1%) or good (GU; *n* = 104, 20.9%). GU smokers rated higher on all the elements of the program evaluation and experienced less decisional conflict, but also displayed a higher intention to quit. After reviewing the DA, the participants on average had a significantly higher intention to use more EBSCIs, in particular in the form of eHealth. Recommendations to make the DA more usable could include tailoring, using video-based information and including value clarification methods. Furthermore, a hybrid variant in which smokers can use the DA independently and with the guidance of a primary care professional could aid both groups in choosing a fitting EBSCI option.

## 1. Introduction

With eight million deaths occurring worldwide as a result of several types of cancer, cardiovascular diseases and respiratory diseases [1], smoking is the most important cause of preventable death [1,2]. In the Netherlands alone, this represents approximately 20,000 mortality cases [3], but also results in other losses for society, such as work loss because of illness or absence and higher healthcare costs [4]. Beyond added costs, smoking is also one of the factors responsible for greater inequality between high socioeconomic status (SES) and low SES households [5], as people from low SES households are more likely to smoke but have fewer material and social resources [6]. Therefore, a decrease in smoking prevalence is a significant goal for the Dutch public health domain [7].

Currently, approximately 20.2% of the Dutch population smokes [8]. Among the less-educated groups, this percentage is higher, at 23.9% [8]. In 2020, 35.6% of all the Dutch smokers made a serious attempt at quitting [8]. However, only 3–5% of the smokers who attempt to quit without help succeed in their first effort [9] and, on average, as many as 30% or more quit attempts if they are unsuccessful for longer than 12 months [10]. To help smokers in their smoking cessation attempts, several evidence-based smoking cessation interventions (EBSCIs) have been developed. EBSCIs are proven to double the likelihood of successful smoking cessation [11].

There are two principal forms of EBSCIs: behavioral and pharmacological support. Behavioral support can include face-to-face counseling by a healthcare professional (HCP) in the GP setting, such as by a general practitioner (GP) or a practice nurse (PN). Other forms of behavioral support outside the GP setting include counseling by a trained stop coach outside the GP setting [12,13,14,15,16,17], tailored online counseling known as eHealth [18,19], telephone counseling [20] and group counseling [21]. Effectiveness rates of behavioral support range between 1 and 3 percent for very brief advice on quitting [13,22] by a GP and from 7 to 14 percent for more extensive forms of counseling, in contrast to unassisted quit attempts [21,23]. Pharmacological support includes nicotine replacement therapy (NRT; e.g., nicotine gum or patches) and pharmacotherapy. NRT has an effectiveness rate of 50–60% if correctly used, in comparison with no treatment or a placebo [24]. For pharmacotherapy, the effectiveness rate ranges from 55 to 77% in comparison with no treatment or a placebo [25,26]. A combination of behavioral and pharmacological support is recommended and required when the smoker wants to be entitled to reimbursement from the health insurer [7,27,28]. In addition to EBSCIs, there is also non-evidence-based cessation assistance, for which no firm evidence base has yet been found. Examples of non-evidence-based cessation assistance include acupuncture, laser therapy and electrostimulation [29]. These options are not usually reimbursed by health insurers. Providing smokers with information and guidance to help them decide which EBSCI would best fit their needs and preferences might increase their involvement in and commitment to their own treatment processes [30,31]. However, only 25–30% of smokers report having used behavioral counseling methods [32,33].

Reaching out to smokers, motivating them to quit and educating them on EBSCIs are difficult to achieve. The primary care setting (PCS) offers an entry point for reaching out to smokers as most people who smoke visit their PCS yearly, often for related complaints such as asthma and COPD [34,35,36]. Within the PCS, practice nurses (PNs) provide the majority of smoking cessation counseling [37] based on the Dutch guideline for smoking cessation care (DGSCC) [27,28]. This guideline is based on an evidence-based counseling method, the minimal intervention smoking cessation strategy (MIS) [12], which is similar to the internationally used 5A protocol of ask, advise, assess, assist, and arrange [38]. However, PNs do not always adhere to these guidelines, particularly the assist and arrange aspects, in which they are asked to provide smoking cessation counseling or refer smokers to other EBSCIs. This might be due to a lack of knowledge or low self-efficacy related to helping their patients make informed decisions [12,27,39]. An overview of effective evidence-based smoking cessation tools might help counselors and smokers decide on the most preferred cessation method [40].

The usability of such an overview among PNs and smoking patients willing to quit smoking has been explored in earlier research, revealing a high appreciation for but low uptake of the materials [41,42]. However, due to a low recruitment rate during this randomized controlled trial which evaluated interventions among smokers recruited via the PCS [41], this study retested the usability of the materials among a larger group of smokers to explore whether the materials with minimal modifications were suitable to be offered as an online intervention. This study explored the perceptions of smokers by using an adapted standalone version of the decision aid (DA) which could be accessed online without the assistance of PNs. To the best of our knowledge, this was the first study conducted with this setup for this particular subject. To explore whether the DA was also suitable for use without the guidance of PNs, the first aim of this study was to assess the overall usability of the standalone version of the DA. To assess the factors that could possibly be associated with smokers’ views on usability, the second aim was to compare groups who differed in their usability score by focusing on their evaluations of the program and their levels of decisional conflict in making a choice. As the main aim of the DA was to increase the use of EBSCIs, the third aim of this study was to explore a possible change in the intention to use EBSCIs during a potential cessation attempt. This was achieved by measuring intention to use EBSCIs before and directly after reviewing the DA.

## 2. Materials and Methods

### 2.1. Study Design and Procedure

A cross-sectional study was carried out in September 2020. Sampling occurred via an online research recruitment agency (www.panelclix.nl, accessed on 27 August 2020), which provided a database with potential participants who at an earlier stage had indicated smoking. The potential participants received information on the study as well as an invitation to immediately enroll in the study. If the potential participants accepted the study invitation, the participants were automatically transferred to the online questionnaire. At the start of the questionnaire, the participants received a brief explanation of the aim of the intervention, followed by the first part of the questionnaire. After filling in the first part of the questionnaire, the participants were asked to review the DA materials, followed by the second part of the questionnaire (all the questions are described below). If they completed the entire questionnaire, they received compensation from the recruitment agency. Inclusion criteria were as follows: (1) the participants were above the age of 18, (2) the participants had smoked (primarily cigarettes) in the past seven days and (3) the participants were able to understand Dutch and had the necessary Internet literacy skills to use the DA. The intention to stop smoking was not an inclusion criterion but the participants had to be willing to consider smoking cessation options. Informed consent was provided prior to the start of the questionnaire by asking if the participants wanted to take part in the study and whether they gave the research team permission to use their data for scientific research.

### 2.2. Materials

The DA was named “StopWijzer”, which can be translated as either “stop-guide” or “stop-smarter”, and it was based on a needs assessment consisting of a literature review (e.g., [18,43,44]), individual semi-structured interviews with general practitioners (GPs) (*n* = 5), practice nurses (PNs) (*n* = 20) and smokers (*n* = 9), a Delphi study on the referral to EBSCIs [27,39] and the input of an advisory board consisting of experts representing various Dutch smoking cessation-related organizations, six of which were actively involved. After the intervention was pilot-tested, the DA was originally deployed to be used in primary care [41,42]. When necessary, the components were reframed to fit the participants’ viewpoints instead of the viewpoint of the PCS. All the materials were written in clear and comprehensible language in accordance with the applicable Dutch guidelines (language level B1 [45]).

In keeping with the DGSCC [27,28], the EBSCIs included in the DA were (1) face-to-face counseling [12], (2) counseling via the Internet (eHealth) [18,19], (3) telephone counseling [20], (4) group counseling [21], (5) pharmacotherapy and (6) nicotine replacement therapy. The participants were strongly recommended to use pharmacotherapy and nicotine replacement therapy only in combination with any form of behavioral counseling, as also described in the DGSCC [27,28].

Use of non-evidence-based methods such as acupuncture and e-cigarettes was also discussed in the DA to address potential questions by smokers about their effectiveness, risks, costs and availability. The DA discouraged use of these non-evidence-based methods and promoted using EBSCIs as suitable alternatives.

#### DA Components

The online DA consisted of the following elements (see also [42]):An introduction, which explained the goals and relevance of the DA and summarized the EBSCIs and the other elements of the DA.An overview of the different EBSCIs in the following order: face-to-face counseling; eHealth; counseling via telephone; group counseling; nicotine replacement therapy; pharmacotherapy; non-evidence-based “cessation” methods of acupuncture [29], laser therapy [46], auriculotherapy [47], hypnosis [48] and e-cigarettes (see Figure 1).An overview of the possible reimbursements of EBSCIs by health insurers with a calculation tool to help patients understand how much money they could save by quitting smoking.The website also contained an overview of the options, which could also be downloaded. The overview listed the EBSCIs mentioned above and gave an outline of their target groups, strengths and weaknesses, effectiveness and costs (see Figure 2).

### 2.3. Measurements

In terms of demographic variables, we asked the participants about their gender (0 = man, 1 = woman), age and highest completed education level (1 = low; 3 = high).

Smoking behavior was measured with two items asking “How many regular cigarettes and/or rolling tobacco do you smoke on average per day?” and “Have you used an e-cigarette in the past 7 days?” (1 = no; 2 = yes, with nicotine; 3 = yes, without nicotine; 4 = yes, but I do not know whether with or without nicotine).

Smoking addiction was measured by using six items of the Fagerström test for nicotine dependence (FTND), such as “Do you smoke more often in the first hours after getting up or do you smoke more often during the other hours of the day?”. The answers were converted to an overall total score in which 0 = not addicted and 10 = highly addicted [49]. The previous quit attempts were measured by asking whether the participant had tried to quit smoking in the past year.

The intention to quit was measured on a five-point Likert scale with one item asking the participants if they had the intention to quit (1 = no, definitely not; 5 = yes, definitely).

Readiness to quit smoking was measured on a six-point scale with one item asking the participants whether they intended to quit smoking within a certain period of time (6 = yes, within the next month; 5 = yes, within 1–3 months; 4 = yes, within 4–6 months; 3 = yes, within 1 year; 2 = yes, within 1–5 years; 1 = yes, but not within the next 5 years) [50,51].

#### 2.3.1. Usability, Program Evaluation and Decisional Conflict

Two items were used to verify whether the participants (1) looked at and (2) read the DA materials (1 = all the materials; 5 = none of the materials).

The usability of the DA was measured by using the system usability scale (SUS) [52] consisting of the sum of 10 items (e.g., “I found the DA complex”), which could be rated on a five-point Likert scale (1 = strongly disagree; 5 = strongly agree), forming a total score from 0 = bad usability to 100 = good usability (Cronbach’s α = 0.66).

The program evaluation was measured with seven constructs of program evaluation as also used in the previous research [53]. Each construct originally consisted of three items measured on a five-point scale (1 = totally disagree; 5 = totally agree). The negatively worded items were reverse-coded. Table 1 summarizes the concepts measured, the example questions and their internal consistency. Based on the unsatisfactory Cronbach’s alpha score, one item was deleted from the comprehension subscale. In addition, the adaptation and dose-inflicted subscales were omitted from the final scale.

The program evaluation was supplemented with one item enquiring whether the participants would recommend the DA to other people willing to stop smoking (1 = totally disagree; 5 = totally agree) and one item asking the participant to rate the overall DA on a scale from one to 10.

Decisional conflict was measured with the decisional conflict scale (DCS) [54,55] consisting of 16 items (e.g., “I feel I have made an informed choice”) on a five-point Likert scale (1 = strongly disagree; 5 = strongly agree). Table 1 summarizes the concepts measured, the example questions and their internal consistency.

#### 2.3.2. Intention to Use EBSCIs

The main goal of the DA was to promote the use of EBSCIs in order to potentially increase their use among smokers undertaking a quit attempt. Therefore, at the start of the questionnaire and directly after reviewing the DA, the participants were asked if they intended to use an EBSCI if they decided to quit smoking. The participants were presented with 10 options (face-to-face with a GP; face-to-face with a PN; face-to-face with a stop coach; eHealth; in groups; via telephone; NRT; pharmacotherapy; non-evidence-based methods; none), to response to which was measured on a dichotomous scale (0 = no; 1 = yes).

### 2.4. Data Analysis

Descriptive statistics were used to describe the characteristics of the recruited participants. The participants were divided into two groups based on their scoring of the usability of the DA using the SUS. As a SUS score above 68 is considered to be above average for web-based interventions [56], this score was used as a cutoff mark between the groups: moderate usability (MU) (mean SUS between 51 and 68) and good usability (GU) (mean SUS above 68). Chi-squared tests and *t*-tests were conducted to test the differences between both groups on program evaluation and decisional conflict after reviewing the DA materials. For the intention to use EBSCIs, the changes were examined before reviewing the materials (pre-test) and after reviewing the materials (post-test), both in the MU group and in the GU group. A paired samples *t*-test was used to test the pre- and post-test difference in the total number of EBSCIs that the participants intended to use. McNemar’s test was used to assess the intention to use individual forms of EBSCIs (yes/no) before and after reviewing the materials. To assess whether the intention to use EBSCIs after reviewing the materials differed significantly between the MU and GU groups, Δ-scores were constructed indicating the differences before and after reviewing the materials. These scores were compared by means of a *t*-test (total number of EBSCIs) and χ^2^ analysis (individual EBSCIs).

## 3. Results

### 3.1. Study Sample Characteristics

The recruitment resulted in 497 participants, most of whom evaluated the DA as moderately usable (MU; *n* = 393; 79.1%). The participants were on average 41 years of age, slightly more often male than female, had mostly a medium-to-high level of education, had a low-to-moderate level of nicotine addiction, smoked an average of 12.5 cigarettes per day and generally did not use e-cigarettes (Table 2). Although both groups indicated readiness to quit, in the group of GU smokers, this intention was significantly higher. However, smokers from the GU group were not significantly more ready to quit, as both groups indicated being ready to quit within six to 12 months on average.

### 3.2. Program Evaluation and Decisional Conflict

Both groups mostly appreciated the DA being comprehensive but expressed least appreciation for the extensive amount of information that the DA contained. Participants from the GU group scored significantly higher on all the factors of the program evaluation scale (*p* < 0.001), indicating that they found the DA more attractive, understandable, suited to their own needs, useful, valuable in making their decision and persuasive, in comparison to the MU group (Table 3). They also found the level of information provided by the DA better dosed than the MU group. Participants from the GU group also indicated significantly more often that they would recommend the DA to others willing to undertake a smoking cessation attempt and gave the DA a significantly higher mark on a scale from one to 10, namely an 8.6 (from good to very good).

Participants from the GU group reported significantly less decisional conflict, both overall and for the subscales, in comparison with participants from the MU group (Table 4). Both groups reported feeling the most conflicted by a feeling of uncertainty (e.g., “I feel sure about what to choose”). For the MU group, their score on this scale exceeded the cutoff point of 37.5, which is associated with decision delay or feeling unsure about implementation [55]. Smokers from the GU group reported being the least conflicted by their level of being informed, but all their scores fell below the cutoff point of 25 [55], indicating that they perceived themselves as having an adequate overview of the options available to them after reviewing the DA materials (60). The MU group of smokers experienced the least conflict about their level of effective decision-making (e.g., “I feel like I have made an informed choice”), although their score did not meet the cutoff point of less than 25, indicating no substantial certainty in their level of decision-making.

### 3.3. Intention to Use EBSCIs

The third aim of this study was to explore a possible change in the intention to use EBSCIs during a potential cessation attempt (see Table 5). Participants in both groups reported an overall and significantly higher intention to use more EBSCIs after they had reviewed the DA materials in comparison to their intention before reviewing the materials. Regarding individual forms of EBSCIs, this difference was specifically significant for their intention to use eHealth. The intention of the participants to not use any EBSCIs when making a quit attempt significantly decreased. No differences were found regarding the usage of non-EBSCIs (NEBSCIs).

Furthermore, participants from the GU group showed a significantly higher increase in the intention to use more EBSCIs eHealth after reviewing the materials in comparison with the MU group.

## 4. Discussion

The aims of this study were to (1) investigate the overall usability, (2) compare groups who rated the DA as having, respectively, moderate and good usability in their evaluation of the program and (3) explore a potential change in the intention to use EBSCIs before and directly after reviewing the DA. Overall, we found that most participants evaluated the usability of the DA as moderate or good. The GU smokers rated higher on all the elements of the program evaluation and experience-led decisional conflict, but also displayed a higher intention to quit. These differences were significant. After reviewing the DA, the participants on average had a significantly higher intention to use more EBSCIs, in particular in the form of eHealth.

With regard to the first objective, the results suggest that most participants found the DA moderately usable in the form in which it was presented, whereas the smokers willing to quit scored the DA’s usability as good. Although both groups had an intention to quit smoking, this intention was significantly higher in the participants from the GU group. A higher intention to quit might also indicate greater interest in the materials, given that according to socio-cognitive models such as the health belief model [57], the theory of planned behavior [58] and the I-change model [43], a person’s beliefs about the effectiveness and perceived benefits—among other factors, such as perceived susceptibility, severity and barriers—might regulate a person’s interest in a behavior change. Furthermore, research has shown that smokers contemplating quitting within the next six months but not within the coming month [59] might benefit the most from information about the intended behavior and from self-efficacy-enhancing information [60]. Therefore, smokers from the GU group may have regarded the information as more relevant to them, which might have resulted in more information retention and absorption and a higher usability score.

The second aim of this study was to compare the groups who scored the DA with moderate and good usability in their evaluation of the program (measured using a program evaluation scale, willingness to recommend the DA to others and scoring the program with an overall mark ranging from 1 to 10) and their level of decisional conflict. Both groups differed in all the aspects, which gives indication of a possible relationship between usability, program evaluation and the DCS; these factors also displayed a significant but moderate correlation in relation to each other. As the DCS measures the perceived conflict in the decision-making process, more conflicted feelings might also regulate the level of usability and appreciation of the DA. Further research is needed to explore the possible relationship between these three concepts in order to provide more in-depth insight into these connections. Both groups found the DA to be comprehensive, although they also indicated that the materials contained an extensive amount of information. Extensive information can be effective for better educated users, such as those in our sample, as they may benefit from the processing of more in-depth information [61]. However, to also reach less-educated groups of users, it is important in stimulating comprehension and attracting attention that this information be made accessible, and these aspects of the DA were less well-rated in this study. Overall, the DA was positively received, with both groups giving it a satisfactory grade.

Regarding decisional conflict, both groups expressed a high level of uncertainty about how to make the actual decision for an EBSCI (e.g., “I feel sure about what is the EBSCI for me”), even though they also reported that they had an adequate overview of the available EBSCIs. This might indicate that even though the participants felt informed about the EBSCIs, they were not sure how to make a balanced decision that aligned with their own preferences. As DAs are designed to aid in the informed decision-making process, they should not only provide all the relevant information on the available options, but also include value clarification exercises or methods (e.g., exercises aimed at helping users evaluate a wide range of options in their own specific contexts) to determine which of the options best fits their needs [40,62,63]. Another explanation for this might be that not all the smokers had the intention to quit at the time of reviewing the materials and did not yet, therefore, think deeply about this part of the decision-making process.

The third aim of this study was to explore a possible change in the intention to use EBSCIs during a potential cessation attempt by measuring the intention before and directly after reviewing the DA. A slight but significant increase was found in the total number of EBSCIs that the participants intended to use. The number of participants willing to use eHealth after reviewing the DA materials also increased. Although systematic research about the (cost-)effectiveness of the existing eHealth interventions is still scarce [64,65], the available studies that report on its effectiveness are positive [18,19,66,67,68,69,70]. The demand for eHealth interventions as found in this study necessitates a greater supply of validated (i.e., evidence-based and effective) eHealth interventions. Furthermore, since there are also numerous Internet interventions available that are not evidence-based [64], the potential establishment of certification with which smokers could recognize validated eHealth interventions might further increase the willingness to use eHealth as this would help smokers in the decision-making process.

The results also indicated a significant decrease in the number of participants who stated that they would make a cessation attempt without the help of EBSCIs. This finding is consistent with the aim of the DA as EBSCIs are proven to double the likelihood of successful smoking cessation [11]. A significant decrease in the intention to use other non-evidence-based smoking cessation interventions such as acupuncture and laser therapy was not found [29]. As research has shown that smokers use NEBSCIs almost as often as they use EBSCIs [71], more attention should be paid to understanding why ineffective methods are still preferred by some smokers and which information they may need to steer them away from these options.

### 4.1. Potential Strengths and Limitations of the Study

One of the strengths of this research was the use of validated questionnaires to measure the relevant constructs. Another strength was the inclusion of a large proportion of smokers who were willing but not yet ready to quit (those in the contemplation phase), in contrast to other studies that usually include self-selected smokers who are ready to make a quit attempt. This factor yielded the added advantage that the smokers were likely not to have sought information on EBSCIs prior to the study or had decided on a form of EBSCI beforehand. However, this also included a limitation as smokers with no intention to quit might look for other information during that phase. However, all the smokers were informed of the aim of the study in advance and were instructed to take on the mindset of someone who is willing to quit smoking within a short period before and after reviewing the materials and during the questionnaire.

The second limitation was that the DA was primarily developed to be used with the aid of a PCP, such as a PN, in the PCS [41,42]. The content of the DA, however, was developed using theoretical grounds based on relevant constructs from the previous studies [18,27,43,44], a needs assessment in the form of a Delphi study [39] and the input of an advisory board. The DA used in this study was adapted by rewriting the materials to fit within the smokers’ frame of reference, taking into account the patients’ potentially low health literacy and rewriting the information using clear and comprehensible language, in accordance with the applicable Dutch guidelines (language level B1) [45].

The third limitation consisted in the use of a cross-sectional design [72] instead of a more longitudinal design, such as a randomized controlled trial, as used in the previous research on the DA materials [41]. Therefore, conclusions on the effectiveness of the intervention in a real-life setting could not be drawn. However, as the main aim of this study was to explore the usability of the materials, this study serves as a pilot test for potential further development of the DA materials.

The last limitation was the use of an online research agency, which resulted in the recruitment of a relatively highly educated participant sample. An additional consequence might be that the participants only took part for the compensation they would receive from this agency and, therefore, did not complete the questionnaire carefully. This was guarded against by including a warning that the participants who did not fill out the questionnaire would not receive a reward. The researchers also screened the data for time of completion and to exclude the participants who fell below the average completion time, but this did not result in the exclusion of any participant.

### 4.2. Practical Implications

As almost 80% of the group of participants rated the materials of the DA only moderately usable, the researchers can cautiously conclude that the materials in their current form are not usable as a standalone DA. To adapt the DA in a way that best fits its potential users, qualitative studies such as read-aloud interviews or pilot groups could aid in pinpointing concrete facilitators and barriers for the usage and reception of the DA. To draw conclusions on the effectiveness of the DA on EBSCI usage and effectiveness, randomized controlled trials conducted as described in similar research [73] are recommended. In order to decrease the amount of information within the DA, the information provided to users could be tailored to their prior knowledge or levels of interest [70,74]. A further communication strategy to also reach a greater number of less-educated smokers might include more video-based information as the previous studies suggested the advantages of using video-based communication over text-based communication [18,75,76]. Last, as the participants in this study indicated that they found it difficult to make a firm decision, the use of value clarification methods could aid in steering the decision-making process by helping users explore their preferences [40,62,77].

Furthermore, based on the findings of this study and their own experiences with the DA in the primary care setting [41], the researchers suggest that the utilization of a hybrid version (i.e., of blended care) that could be used both within the PCS and as a standalone option could be a feasible option for further development of the DA. As mentioned above, PCPs in the PCS work with the DGSCC [27,28] based on the 5A protocol (i.e., ask, advise, assess, assist and arrange) [38]. However, as the time within the PCS is very limited, an abbreviated version of this protocol has been proposed, the ask–advise–refer (AAR) strategy [22], which can be used to structure a very brief advice by a PCP and has already been successfully used in Dutch cardiac wards [78]. PCPs can use the DA as a reference during the referral part of this strategy, while smokers can use the online materials to further explore the available EBSCI options after their consultation with a PCP. Another advantage of adapting the DA into a hybrid variant is that it may benefit from the Internet interventions’ broad reach but could also have the advantage of the higher adherence rate of interventions used in healthcare settings [79]. Another advantage of the so-called blended care is that it allows the combination of personal attention and synchronous communication with the online advantages of high accessibility [80,81]. Given that the primary care setting prominently reaches smokers who are more motivated to quit [79,82], using a mass media approach might reach a greater absolute number of smokers, even those who are still in the (pre-)contemplating phase [79], as was also the case in this study.

## 5. Conclusions

As the use of EBSCIs can double the likelihood of a successful smoking cessation attempt, this study investigated the usability of a DA aimed at increasing the use of EBSCIs. As the DA was originally designed to be used in general practice with the guidance of a PN, the aim of this study was to explore the usability of an adapted standalone version of the aid among a large group of smokers. Most participants found the DA only moderately usable, although those who intended to quit found it more usable. The participants who found the usability of the DA to be good rated higher on all the elements concerning the evaluation of the DA, including the recommendation to others and the overall mark, and experienced less decisional conflict. Furthermore, after reviewing the DA, the participants on average had a significantly higher intention to use more EBSCIs, in particular in the form of eHealth. Recommendations to make the DA more usable and well-received among a broader group of smokers could include tailoring, transforming text-based information into video-based information and including value clarification methods. Furthermore, as the DA was found to be only moderately usable in the standalone version, a hybrid variant that would allow smokers to use the DA both on their own and with the guidance of PCPs could aid both groups in choosing a fitting EBSCI option.

## Figures and Tables

**Figure 1 ijerph-18-10836-f001:**
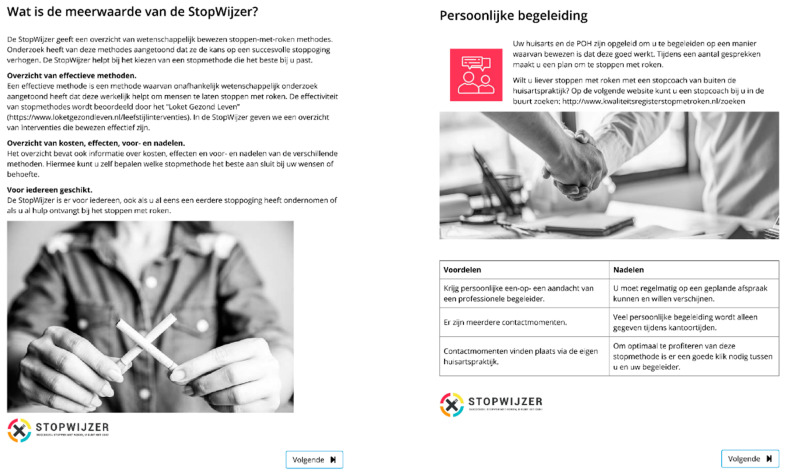
Excerpts from the DA website.

**Figure 2 ijerph-18-10836-f002:**
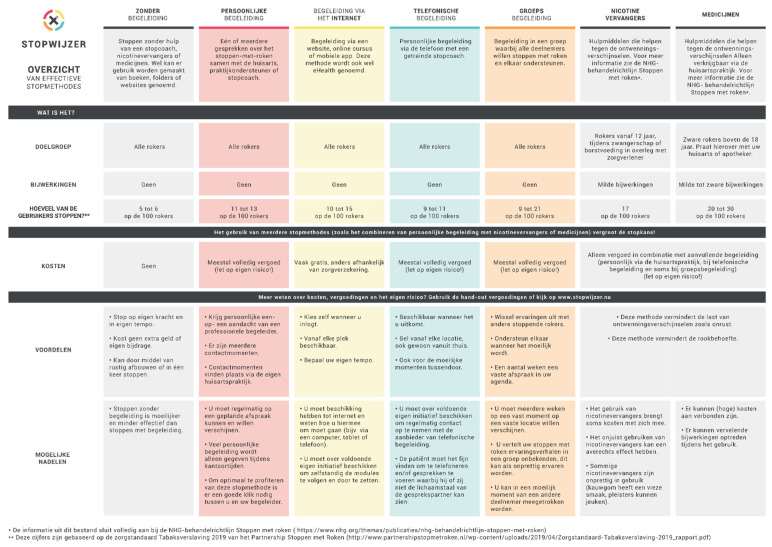
Decision overview (option grid).

**Table 1 ijerph-18-10836-t001:** Constructs of the program evaluation scale and decisional conflict scale.

	Example Questions	Cronbach’s α
Program evaluation scale constructs		
Attention	The DA held my attention	0.81
Comprehension	In my opinion, the DA is clear	0.81 ^1^
Adaptation	The DA applied to me personally	0.44 ^2^
Appreciation	The DA is interesting	0.81
Processing	The DA contains good tips on the best way to quit smoking	0.87
Dose infliction	The DA provides a nice overview of the available evidence-based smoking cessation methods	0.46 ^2^
Persuasion	The DA is convincing	0.80
Complete scale	–	0.93
Decisional conflict scale constructs		
	By using the DA, …	
Uncertainty	I know what the best choice is for me	0.84
Informed	I know which options are available to me	0.85
Value clarity	I am clear about which benefits matter the most to me	0.84
Support	I have enough support to make a choice	0.75
Effective decision	I am satisfied with my choice	0.78
Complete scale	–	0.94

^1^ With one item deleted from the scale. ^2^ This subscale was omitted from the total program evaluation scale.

**Table 2 ijerph-18-10836-t002:** Characteristics of the sample including smoking behavior.

Study Sample Characteristics	Total (*n* = 497)	MU (*n* = 393)	GU (*n* = 104)	*t*	χ^2^	*p*
Age (years), mean (SD)	41.23 (13.9)	41.06 (13.9)	41.96 (12.6)	−0.597		0.551
Female, *n* (%)	225 (45.3)	172 (43.8)	53 (51.5)		1.947	0.163
Educational level, *n* (%)					0.174	0.916
Low	59 (11.9)	47 (12)	11 (10.7)			
Medium	229 (46.1)	180 (45.8)	49 (47.6)			
High	209 (42.1)	166 (42.2)	43 (41.7)			
FTND score ^1^, mean (SD)	4.24 (2.4)	4.32 (2.5)	3.94 (2.4)	1.412		0.159
Number of cigarettes smoked/day, mean (SD)	12.51 (7.7)	12.56 (7.8)	12.38 (7.1)	0.215		0.830
Use of e-cigarettes, *n* (%)					4.421	0.219
No	306 (61.6)	246 (62.6)	59 (57.3)			
Yes, without nicotine	40 (8.0)	35 (8.9)	5 (4.9)			
Yes, with nicotine	144 (29.0)	107 (27.2)	37 (35.9)			
Yes, do not know whether with or without nicotine	7 (1.4)	5 (1.3)	2 (1.9)			
Previous quit attempt undertaken, *n* (%)	309 (62.2)	248 (63.1)	61 (59.2)		0.414	0.520
Intention to quit ^2^	3.97 (0.9)	3.88 (0.9)	4.29 (0.8)	−4.334		0.000
Readiness to quit ^3^	3.19 (1.3)	3.20 (1.9)	3.12 (1.4)	0.579		0.563

^1^ Range from 1 to 10, 0 = not addicted; 10 = highly addicted. ^2^ 1 = no, definitely not; 5 = yes, definitely. ^3^ 1 = yes, not within the next 5 years; 6 = yes, within the next month.

**Table 3 ijerph-18-10836-t003:** Comparison of the mean scores on usability, program evaluation, recommendation to others and grading mark of MU and GU smokers.

	Total (*n* = 497)	MU (*n* = 393)	GU (*n* = 104)	*t*	*p*
Program evaluation scale ^1^	2.42 (0.4)	3.47 (0.6)	4.27 (0.5)	−12.674	0.000
Attention subscale	3.47 (0.8)	3.30 (0.8)	4.12 (0.7)	−9.835	0.000
Comprehension subscale	3.94 (0.7)	3.77 (0.7)	4.59 (0.6)	−11.301	0.000
Comprehension: difficult	3.79 (1.0)	3.60 (0.9)	4.53 (0.8)	−9.191	0.000
Adaptation: fitted situation	3.45 (0.9)	3.32 (0.9)	3.97 (0.8)	−6.581	0.000
Adaptation: lacked information	3.19 (1.0)	3.05 (0.9)	3.72 (1.0)	−6.605	0.000
Adaptation: too general	3.50 (1.0)	3.36 (0.9)	4.05 (1.0)	−6.435	0.000
Appreciation subscale	3.59 (0.8)	3.43 (0.8)	4.20 (0.6)	−9.386	0.000
Process subscale	3.53 (0.8)	3.37 (0.7)	4.16 (0.6)	−9.741	0.000
Dose subscale	3.76 (0.8)	3.59 (0.8)	4.46 (0.5)	−11.126	0.000
Dose: much information	3.77 (0.8)	2.69 (1.0)	3.57 (1.2)	−7.518	0.000
Persuasion subscale	3.74 (0.7)	3.57 (0.7)	4.38 (0.5)	−9.051	0.000
Recommendation ^2^	3.75 (0.9)	3.55 (0.8)	4.52 (0.6)	−10.606	0.000
Mark (1–10)		7.27 (1.3)	8.56 (0.9)	−11.531	0.000

^1^ 1 = totally disagree, 5 = totally agree. ^2^ 1 = would not recommend, 5 = would recommend.

**Table 4 ijerph-18-10836-t004:** Comparison of the mean scores on decisional conflict of the MU and GU smokers.

	Total (*n* = 497)	MU (*n* = 393)	GU (*n* = 104)	*t*	*p*
Decisional conflict scale, mean (SD) ^1^	31.73 (14.8)	35.56 (13.3)	17.13 (10.6)	13.060	0.000
Uncertainty subscale	34.17 (18.0)	38.13 (16.7)	19.09 (14.2)	10.583	0.000
Informed subscale	29.60 (17.9)	34.01 (16.2)	12.78 (13.5)	12.224	0.000
Value clarity subscale	32.88 (17.9)	36.70 (16.4)	18.28 (15.7)	10.218	0.000
Support subscale	32.34 (17.6)	36.28 (16.7)	17.31 (12.3)	10.792	0.000
Effective decision subscale	30.18 (15.0)	33.40 (13.9)	17.90 (12.7)	10.255	0.000

^1^ 5 = no decisional conflict, 100 = a lot of decisional conflict.

**Table 5 ijerph-18-10836-t005:** Comparison of the intention to use EBSCIs measured before and after reviewing the DA.

	MU (*n* = 393)		GU (*n* = 104)		Comparison of Changes between the MU Group and the GU Group
	Before	After	Before	After	
Intention to use EBSCIs, mean amount (SD) ^1^	1.47 (1.1)	1.59 (1.1) *	1.89 (1.4)	1.91 (1.1)	NS
Behavioral counseling, % (n)					
GP	10.2 (40)	9.9 (39)	17.5 (18)	17.5 (18)	NS
PN	15.8 (62)	18.3 (72)	24.3 (25)	23.3 (24)	NS
Stop coach	12.7 (50)	16.0 (63)	15.5 (16)	15.5 (16)	NS
eHealth	9.9 (39)	16.3 (64) **	12.6 (13)	30.1 (31) **	Δ GU > Δ MU *
In groups	7.1 (28)	8.1 (32)	1.9 (2)	4.9 (5)	NS
Via telephone	7.6 (30)	9.9 (39)	6.8 (7)	10.7 (11)	NS
NRT	25.2 (99)	24.9 (98)	47.6 (49)	40.8 (42)	NS
Pharmacotherapy	13.0 (51)	15.8 (62)	25.2 (26)	23.3 (24)	NS
NEBSCI ^2^	8.7 (34)	7.9 (31)	11.7 (12)	8.7 (9)	NS
None	37.2 (146)	31.8 (125) **	26.2 (27)	16.5 (17) *	NS

^1^ One category excluded. ^2^ For example, acupuncture, hypnotherapy or laser therapy. * *p* < 0.01, ** *p* < 0.001.

## Data Availability

Data can be accessed upon reasonable request to the corresponding author (d.zijlstra@maastrichtuniversity.nl).

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
