# Peer review of "The Usability of an Online Tool to Promote the Use of Evidence-Based Smoking Cessation Interventions"

_ijerph, 2021, doi:10.3390/ijerph182010836_

Round 1
Reviewer 1 Report
This is an interesting study on the significant tobacco control problems. The manuscript is well-prepared.
Please consider the following changes:
- Please clearly define sampling methods and how the participants were recruited
- Please provide 2-3 sentences on the development of DA components
- The first paragraph of the discussion section should be revised. Usually, this paragraph should briefly summarize the main findings rather than repeat the study aim
Author Response
This is an interesting study on the significant tobacco control problems. The manuscript is well-prepared.
Thank you for your compliments.
Please consider the following changes:
- Please clearly define sampling methods and how the participants were recruited
Based on your feedback we have reformulated our sampling and recruitment method.
‘A cross-sectional study was conducted in September 2020. Sampling took place via an online research recruitment agency (www.panelclix.nl), which provided a database with potential participants who had previously indicated that they smoke. Potential participants received information about the study and an invitation to take part in the study. If potential participants accepted the survey invitation, they were automatically directed to the online questionnaire.’
- Please provide 2-3 sentences on the development of DA components
We have added the following sentences to the method section on the development of DA components:
‘Where necessary, components were adapted to the point of view of the participants rather than that of a PCS. All materials were written in clear and understandable language, in accordance with applicable Dutch guidelines (Language level B1 - [44])’
- The first paragraph of the discussion section should be revised. Usually, this paragraph should briefly summarize the main findings rather than repeat the study aim
We have added the following sentences to the first paragraph of the introduction:
‘Overall, we found that that most participants evaluated the usability of the DA as moderate or good. GU smokers rated higher on all elements of the program evaluation and experiences led decisional conflict but also displayed a higher intention to quit. These differences were significant. After reviewing the DA, participants on average had a significantly higher intention to use more EBSCIs, in particular in the form of eHealth.’
Reviewer 2 Report
It is an interesting manuscript concerning smoking cessation interventions, which is an extremely important issue. This study is well designed and clearly described, however some improvement would be appreciated. In the introduction section I have not found information if similar studies has already beed conducted and this should be mentioned. If there are similar studies, they should be briefly described in the discussion section and compared with the obtained results. I also suggest to add graphic presentation of the results. Some spelling and editorial mistakes should also be corrected.
Author Response
It is an interesting manuscript concerning smoking cessation interventions, which is an extremely important issue. This study is well designed and clearly described, however some improvement would be appreciated.
Thank you for your compliments.
In the introduction section I have not found information if similar studies have already been conducted and this should be mentioned. If there are similar studies, they should be briefly described in the discussion section and compared with the obtained results.
Studies that are related to this study (e.g., (Stanczyk et al., 2014; Stanczyk et al., 2011)) are mentioned in the introduction and discussion. We added the following sentence to the introduction:
‘To our best knowledge, this was the first study conducted with this setup for this particular subject.’
I also suggest to add graphic presentation of the results.
We are unsure how we can comply with this suggestion. Usually, journals prefer tables. If necessary we can translate the tables into bars or infographics, but we believe that this decision should be left to the editor, as these are journal preferences and styles.
Some spelling and editorial mistakes should also be corrected.
The manuscript was checked by a native English speaker before submission. Following your suggestion, we have thoroughly checked the revised manuscript to correct any remaining mistakes. Corrections were made using the track-changes function in the manuscript file.
References
Stanczyk, N., Bolman, C., van Adrichem, M., Candel, M., Muris, J., & de Vries, H. (2014). Comparison of text and video computer-tailored interventions for smoking cessation: randomized controlled trial. J Med Internet Res, 16(3), e69. https://doi.org/10.2196/jmir.3016
Stanczyk, N. E., Bolman, C., Muris, J. W., & de Vries, H. (2011). Study protocol of a Dutch smoking cessation e-health program. BMC Public Health, 11(1), 847. https://doi.org/10.1186/1471-2458-11-847